# An unexpected N-terminal loop in PD-1 dominates binding by nivolumab

Shuguang Tan[1,2,*], Hao Zhang[1,3,*], Yan Chai[2,*], Hao Song[4], Zhou Tong[1], Qihui Wang[1], Jianxun Qi[2], Gary Wong[2], Xiaodong Zhu[5], William J. Liu[6], Shan Gao[7], Zhongfu Wang[8], Yi Shi[2], Fuquan Yang[9], George F. Gao[2,4,6] & Jinghua Yan[1,2,3]

Cancer immunotherapy by targeting of immune checkpoint molecules has been a research 'hot-spot' in recent years. Nivolumab, a human monoclonal antibody targeting PD-1, has been widely used clinically since 2014. However, the binding mechanism of nivolumab to PD-1 has not yet been shown, despite a recent report describing the complex structure of pembrolizumab/PD-1. It has previously been speculated that PD-1 glycosylation is involved in nivolumab recognition. Here we report the complex structure of nivolumab with PD-1 and evaluate the effects of PD-1 N-glycosylation on the interactions with nivolumab. Structural and functional analyses unexpectedly reveal an N-terminal loop outside the IgV domain of PD-1. This loop is not involved in recognition of PD-L1 but dominates binding to nivolumab, whereas N-glycosylation is not involved in binding at all. Nivolumab binds to a completely different area than pembrolizumab. These results provide the basis for the design of future inhibitory molecules targeting PD-1.

[1] CAS Key Laboratory of Microbial Physiological and Metabolic engineering, Institute of Microbiology, Chinese Academy of Sciences, Beijing 100101, China. [2] CAS Key Laboratory of Pathogenic Microbiology and Immunology, Institute of Microbiology, Chinese Academy of Sciences, Beijing 100101, China. [3] School of Life Sciences, Anhui University, Hefei 230601, China. [4] Research Network of Immunity and Health (RNIH), Beijing Institutes of Life Science, Chinese Academy of Sciences, Beijing 100101, China. [5] Beijing Combio Co., Ltd., Beijing 100111, China. [6] National Institute for Viral Disease Control and Prevention, Chinese Center for Disease Control and Prevention (China CDC), Beijing 102206, China. [7] CAS Key Laboratory of Bio-medical Diagnostics, Suzhou Institute of Biomedical Engineering and Technology, Chinese Academy of Sciences, Suzhou 215163, China. [8] Key Laboratory of Resource Biology and Biotechnology in Western China, Ministry of Education and Provincial Key Laboratory of Biotechnology, College of Life Sciences, Northwest University, Xi'an 710069, China. [9] Laboratory of Protein and Peptide Pharmaceuticals & Laboratory of Proteomics, Institute of Biophysics, Chinese Academy of Sciences, Beijing 100101, China. * These authors contributed equally to this work. Correspondence and requests for materials should be addressed to G.F.G. (gaof@im.ac.cn) or to J.Y. (email:yanjh@im.ac.cn).

Immune checkpoint blockade therapy has drawn considerable interest as tumour immunotherapy was selected as Break-through of the Year by Science in 2013 (ref. 1). T-cell activation involves multiple paired molecular interactions including T-cell receptor (TCR)/peptide major histocompatibility complex interactions, CD4 (or CD8)/peptide major histocompatibility complex co-receptor interactions and co-stimulatory ligand-receptor interactions in the two-signal system[2–7]. Furthermore, activated T cells also require co-stimulatory and co-inhibitory molecules to modulate TCR-mediated, antigen-specific T-cell responses and self-tolerance[8–10]. A proper balance between activating and inhibitory immune molecule functions is pivotal for a robust immune response.

As a member of the CD28 superfamily molecule, programmed cell death 1 (PD-1) was first discovered as a gene upregulated in a T-cell hybridoma undergoing cell death[11]. PD-1 ligand 1 (PD-L1) and then PD-1 ligand 2 (PD-L2) were subsequently identified to be the ligands of PD-1 (refs 12,13). Studies show that upregulation of PD-1 and PD-L1 expression in T cells and tumour cells, respectively, would induce immune suppression in the tumour microenvironment, an important mechanism for tumour immune escape[14,15]. Indeed, PD-1/PD-L1 paired signalling is important in monoclonal antibody (MAb)-based immune checkpoint blockade therapy, based on multiple evidence indicating that restoration of anti-tumour immune responses and the inhibition of intratumoral Treg cells within the tumour microenvironment could be achieved by the blockade of the PD-1 pathway[8,16–18]. Therefore, MAbs targeting PD-1 or PD-L1 are a priority for the development of tumour immuno-therapeutics[19–21].

As 2014, two MAbs targeting PD-1 have been approved by the US Food and Drug Administration for treating melanoma, non-small cell lung cancer and head and neck squamous cell carcinoma, the pembrolizumab from Merck & Co., Inc. (Kenilworth, USA) and the nivolumab from Bristol-Myers Squibb (New York, USA)[9]. Clinical application of nivolumab has shown promising tumour suppressive activity in multiple tumours, including melanoma, metastatic renal cell carcinomas and non-small cell lung cancer[22–24], with an overall objective response rate of 30-40% in melanoma patients[23,25]. Complex structures of MAb drugs and their targets have been determined recently by Na et al. (2016) and our group, the anti-PD-1 pembrolizumab complexed with human PD-1 (hPD-1) and the anti-PD-L1 avelumab complexed with human PD-L1 (hPD-L1), which reveals the molecular basis for antibody based immuno-therapy for tumours[26–29]. The recently reported nonpeptidic chemical inhibitor targeting PD-L1 suggests that there are 'hot spots' on PD-L1 for the design of PD-L1 antagonist drugs[30]. However, the structural basis of PD-1/PD-L1 blockade by nivolumab remains unclear. It is unknown whether there are some 'hot spots' on PD-1 for MAb interaction, or whether nivolumab utilizes a similar or distinct competitive binding mode compared with pembrolizumab to PD-1. These key scientific questions need to be addressed, as they are important for the development of future blockers/inhibitors, esp. small molecule inhibitors and next-generation MAbs.

Protein glycosylation is an important post-translational modification, which has a key role in many important biological processes[31]. O-linked and N-linked glycosylations are two of the most common mechanisms for linking glycans to proteins[32]. Alterations in glycosylation patterns, resulting from incomplete synthesis and neo-synthesis processes in tumour cells, were shown to be important in differential processing following malignant transformation[33,34]. Tumour cells display numerous alterations in glycosylation patterns compared with non-malignant cells, with altered tumour cell functions in cell adhesion, migration, interactions with the cell matrix, immune surveillance and cellular metabolism among others[35]. Recent studies revealed that PD-1 expression was also observed in cancer cells and the blockade of the intrinsic PD-1 with MAbs would suppress tumour growth[36]. PD-1 has four predicted N-linked glycosylation sites within its extracellular immunoglobulin variable (IgV) domain (Fig. 1a). A preclinical study previously demonstrated that the binding of nivolumab to PD-1 likely depends on the glycosylation of PD-1, as nivolumab bound only glycosylated PD-1 expressed in a mammalian cell line but not non-glycosylated PD-1 expressed in E. coli. (http://www.accessdata.fda.gov/drugsatfda_docs/nda/2014/125554Orig1s000PharmR.pdf). This implies that possible alterations in PD-1 glycosylation patterns in tumour cells would affect the interaction with PD-1-targeting MAbs and subsequently influence the efficacy of immune checkpoint blockade therapy. Therefore, the influence of PD-1 glyco-sylation on immune checkpoint blockade MAbs needs to be investigated and nivolumab constitutes a good model in this regard.

In the present study, we report the structural basis of nivolumab binding to PD-1. Furthermore, we analyse the glycosylation modification of PD-1 and the resultant effects on nivolumab/PD-1 interactions. We find that four predicted potential N-linked glycosylation sites exist in the PD-1 IgV domain. The binding of nivolumab to PD-1 is not glycosylation-dependent, rather an N-terminal loop (N-loop) of PD-1 domi-nates binding, providing an unexpected target for the develop-ment of future PD-1/PD-L1 blockade drugs. The binding epitopes of nivolumab to PD-1 are completely different from pembroli-zumab, another clinically used MAb from which the complex structure is known.

## Results

**Overall structure of the PD-1 nivolumab-Fab complex.** PD-1 is a type I transmembrane protein and the ectodomain consists of the signal peptide, an N-loop, an IgV domain and a stalk region, with four potential N-glycosylation sites within the IgV domain (Fig. 1a). Since it has been previously described that nivolumab cannot bind to refolded hPD-1 protein, we expressed the ectodomain of PD-1 (amino acids 1–167) in the mammalian cell expression system to obtain fully glycosylated PD-1 protein. A binding assay showed that the mammalian-expressed PD-1 protein and nivolumab-Fab are stable during gel filtration (Fig. 1b,c). The PD-1/nivolumab-Fab complex protein was subsequently prepared for crystal screening and the complex structure of PD-1/nivolumab-Fab was solved at a resolution of 2.4 Å (Table 1). The overall structure revealed that nivolumab utilizes both heavy chain ($V_H$) and light chain ($V_L$) to interact with PD-1 with a buried surface of 1932.5 Å[2]. The interaction with PD-1 involved all three CDR loops in the nivolumab $V_H$ region, and CDR2 and CDR1 loops in $V_L$ provided partial interactions, and LCDR3 does not have any contact (Fig. 1d). Fragment (L25–L142) of PD-1 was observed in the structure and N-terminal amino-acid sequencing analysis revealed that the first amino acid of the mature protein was L25 (Supplementary Fig. 1). The binding of nivolumab to PD-1 covered residues in the N-loop, as well as the FG and BC loops of the IgV domain (Fig. 1d). All previous structural studies including both the hPD-1 and murine PD-1 (mPD-1) were focused on the IgV domain[37–39] and this is the first time that a structure with the N-loop was observed.

**PD-1 N-loop dominates interaction with nivolumab.** It is worth noting that the N-loop (L25-P34) of PD-1, which was neglected

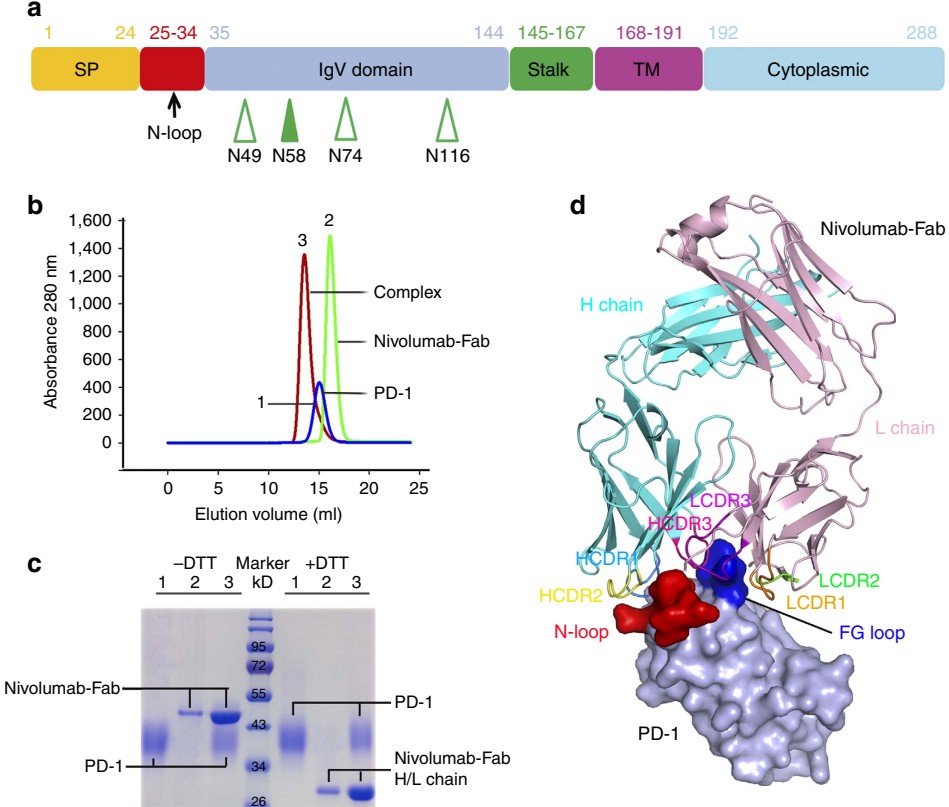

**Figure 1 | Overall structure of PD-1 nivolumab-Fab complex. (a)** Schematic diagram of PD-1 protein. Full-length PD-1 is separated into ectodomain, transmembrane domain and intracellular domain. Ectodomain is composed of the signal peptide, N-loop, IgV domain and stalk region. The N-glycosylation at N58, which could be seen in the complex structure in the ectodomain is indicated with solid green arrowhead, whereas the other three N-glycosylation sites are indicated with hollow green arrowheads. The numbers indicate amino acid positions. **(b)** Gel filtration profiles of PD-1 (blue), nivolumab-Fab (green) and the PD-1-nivolumab-Fab complex (red) were analysed by size-exclusion chromatography as indicated. **(c)** The separation profiles of each pooled samples on SDS-PAGE are shown in reducing ( + DTT) or non-reducing (-DTT) conditions, 1 for PD-1, 2 for nivolumab-Fab and 3 for PD-1-nivolumab-Fab complex. **(d)** The complex structure of nivolumab-Fab bound to PD-1. The Fab fragment of nivolumab is shown as cartoon (Heavy chain, cyan; Light chain, pink), and PD-1 is shown as surface representation (light blue). The CDR1, CDR2 and CDR3 loops of the heavy chain (HCDR1, HCDR2, HCDR3) are coloured in marine, yellow and magenta, respectively. The CDR1, CDR2 and CDR3 loops of light chain (LCDR1, LCDR2, LCDR3) are coloured in orange, violet and green, respectively. The N-loop and FG loop of the PD-1 molecule are highlighted in red and blue, respectively.

in previous studies, was observed for the first time in the complex structure stabilized by nivolumab (Fig. 1d). Structural analysis revealed that the N-loop of PD-1 contributes the majority of hydrogen bonds (10 out of 16) within the nivolumab and PD-1 interface. Specifically, amino acids of the N-loop (L25, S27, P28, D29 and R30) formed 10 hydrogen bonds with those in HCDR1 (S30, N31 and G33) and HCDR2 (W52, Y53 and K57) of nivolumab (Fig. 2a, Supplementary Fig. 2a and Table 1). In addition, the FG loop and BC loop of PD-1 contributed the remaining contacts with nivolumab (Fig. 2b). The FG loop of PD-1 also formed five hydrogen bonds with HCDR3 (D100 and D101) and fragment regions around LCDR2 (Y49 and T56) (Fig. 2b and Supplementary Fig. 2b). The BC loop (T59) of PD-1 also contributed one hydrogen bond interaction with HCDR1 (N31) (Fig. 2a). Taken together, the three loops provided a flexible platform for nivolumab binding. Of note, the N-loop, rather than the IgV domain of PD-1, dominated the interaction with nivolumab.

**N-glycosylation of PD-1 and nivolumab recognition.** Glycosylation, especially N-linked glycosylation, is known to have a critical role in protein folding and function[40,41]. There are four potential N-linked glycosylation sites (N49, N58, N74 and N116)

in the IgV domain of PD-1 (Figs 1a and 3a). Notably, N58 of PD-1, the only glycosylation site near the interface of PD-1-nivolumab-Fab complex, is glycosylated with three glycan rings in the unambiguous electron density (Fig. 3b). N′glycans on N58 consists of two N′acetylglucosamines and one fucose (Fig. 3b). However, the N-glycans do not interact with nivolumab. We observed that the mammalian-expressed PD-1 protein has a much larger molecular weight than the proteins expressed in *Escherichia coli* or insect cells, indicating that the PD-1 protein is heavily glycosylated (Fig. 3c). Alanine scanning of the four glycosylation sites (N49, N58A, N74A or N116A) led to a substantial reduction of the molecular weight compared with wild-type (WT) PD-1, confirming the existence of N-glycosylation at these sites (Fig. 3c).

We then investigated the effects of N-glycosylation on nivolumab binding by using the alanine scanning mutants. We expressed full-length WT or mutated PD-1 protein in 293 T cells, incubated the samples with nivolumab and analysed them with FACS (Fig. 4a). The results showed that the four N-glycosylation site mutant scan still bind to nivolumab. We further evaluated the binding affinities of the WT or site-mutated PD-1 proteins to nivolumab using surface plasmon resonance (SPR) (Fig. 4b). The SPR results demonstrated that the site-mutated PD-1 proteins had similar binding affinities as the WT (Kd ranging

from 1.45 to 3.48 nM). Moreover, we expressed the ectodomain of PD-1(L25-R147) in *E. coli* to evaluate whether post-translational glycosylation affects PD-1-nivolumab interaction. The SPR analysis revealed that the binding affinity with the refolded PD-1 (Kd = 4.03 nM) was comparable to that of PD-1 from mammalian cells (Kd = 1.45 nM) (Fig. 4b). Based on these findings, we concluded that the binding of nivolumab to PD-1 was independent of N-glycosylation, which is different from previous reports(http://www.accessdata.fda.gov/drugsatfda_docs/nda/2014/125554Orig1s000PharmR.pdf).

In order to assess the effect of the N-loop on nivolumab binding, we generated N-loop truncated PD-1 constructs (N33-R147, IgV domain), which were expressed in both *E. coli* and insect cells. The results showed that no binding to the truncated PD-1 proteins was detected for nivolumab (Fig. 4b). On

the other hand, the binding affinity of the N-loop truncated PD-1 (IgV) proteins(*E. coli* refolded, Kd = 4.06 nM, or insect cell expressed, Kd = 2.49 nM) showed no substantial differences to that of the N-loop covered PD-1 (*E. coli* refolded, Kd = 8.43 nM, or mammalian cell expressed, Kd = 6.21 nM) when binding to PD-L1 (Fig. 4b).

**Nivolumab competitive binding with PD-1 ligand PD-L1.** Superposition of the PD-1-nivolumab-Fab complex structure with the PD-1-PD-L1 complex structure (PDB:4ZQK) revealed competitive binding of nivolumab with PD-L1 (Fig. 5a and Supplementary Fig. 3). The interaction of PD-1 and PD-L1 involves both front β-sheet faces of their IgV domains, whereas the binding of PD-1 to nivolumab was dominated by the N-loop accompanied with contributions from both the FG and BC loops (Fig. 5a). The competitive binding of nivolumab relies on $V_L$, which has an overlapping binding surface with PD-L1 (Fig. 5b). The detailed analysis of the buried surface on PD-1 reveals that the overlapped binding area of nivolumab and PD-L1 is mainly located on the FG loop (Fig. 5c). These results indicated that the blockade mechanism of nivolumab is that the $V_L$ of nivolumab provides steric clash to abrogate the binding of PD-L1 to PD-1.

**Distinct blockade binding mode compared with pembrolizumab.** To date, there are only two PD-1-MAb complex structures available including the one reported in this study. A comparative study between nivolumab and pembrolizumab was conducted to elucidate the binding modes of these two clinically approved MAbs (Fig. 6a). The results showed that the two antibodies bind PD-1 in two different orientations with steric clash (Fig. 6a).The binding surface of nivolumab on PD-1 is close to that of pembrolizumab, but they do not overlap (Fig. 6b and Supplementary Fig. 3). We then analysed whether nivolumab showed any additional binding to pembrolizumab saturated PD-1. Only partial complementary binding to the preexisting PD-1-pembrolizumab by nivolumab was detected, and vice versa (Fig. 6c). Hence, though two MAbs touch different PD-1 surfaces, they have steric clash when bound to PD-1.

**Targeting the PD-1 loops for immune checkpoint blockade.** The PD-1-nivolumab-Fab complex structure reported here enabled us to compare the PD-1 expressed in mammalian cells versus that from prokaryotic cells for the first time. Overall, the β-sheet-constituted IgV domain core showed no substantial differences to that expressed in prokaryotic cells, regardless of unbound and bound states. The most obvious differences were observed on the loops, for example, N-loop, FG loop, C'D loop

**Table 1 | Data collection and refinement statistics.**

| Data collection | |
| --- | --- |
| Space group | P43 21 2 |
| Wavelength (Å) | 0.97907 |
| Unit cell dimensions | |
| a, b, c (Å) | 108.527, 108.527, 145.423 |
| α, β, γ (°) | 90.00, 90.00, 90.00 |
| Resolution (Å) | 50.00–2.40 (2.49–2.40)* |
| Observed reflections | 751908 |
| Completeness (%) | 100.0 (100.0) |
| Redundancy | 21.7 (22.1) |
| $R_{merge}$ (%) | 11.4 (69.1) |
| I/σ | 32.1 (6.2) |
| | |
| *Refinement* | |
| $R_{work}/R_{free}$(%) | 18.6/23.6 |
| No. atoms | |
| Protein | 4087 |
| Ligands | 38 |
| Water | 263 |
| *B*-factors | |
| Protein | 42.77 |
| Ligands | 82.39 |
| Water | 42.73 |
| r.m.s. deviation | |
| Bond lengths (Å) | 0.011 |
| Bond angles (°) | 1.20 |
| Ramachandran plot | |
| Favoured (%) | 96 |
| Allowed (%) | 3.8 |
| Outliers (%) | 0.0 |

*Values in parentheses are given for the highest resolution shell.

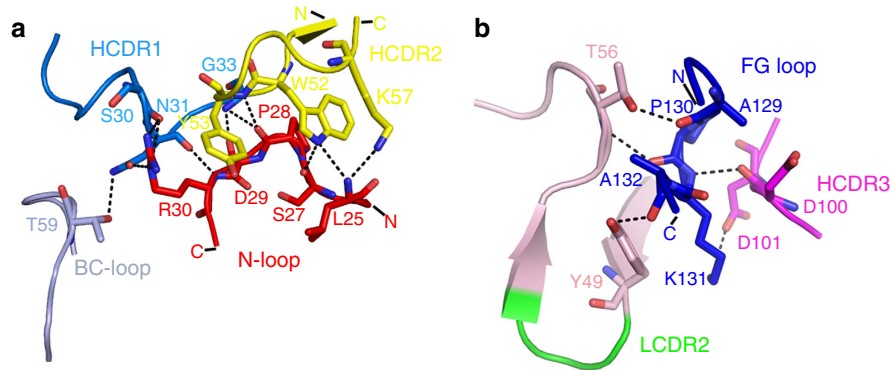

**Figure 2 | The atomic interaction details at the binding interface of PD-1 nivolumab-Fab complex.** Detailed interactions of nivolumab binding to the N-loop, BC loop (**a**) and FG loop (**b**) of PD-1. Residues involved in the hydrogen bond interaction are shown as sticks and labelled. Hydrogen bonds are shown as dashed black lines.

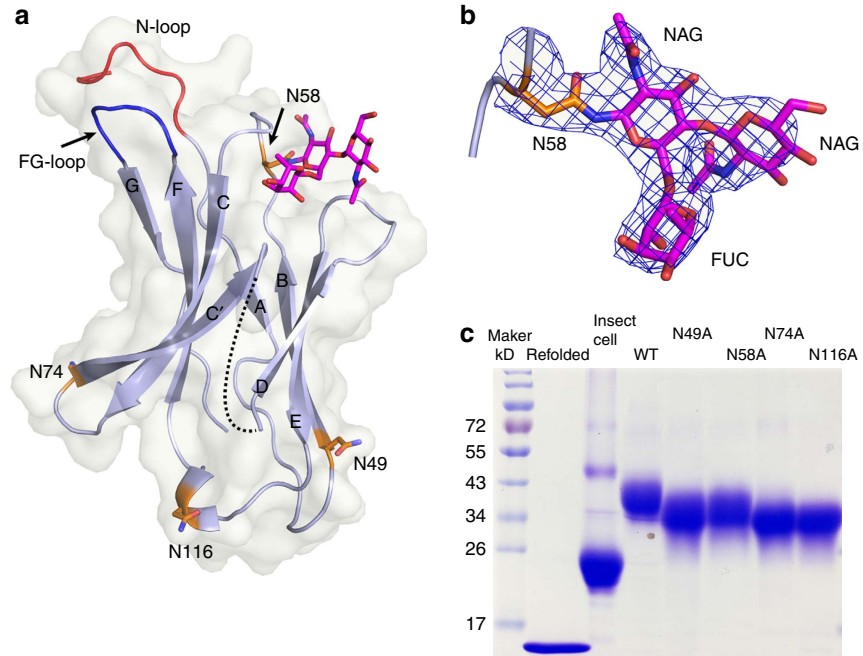

**Figure 3 | Structural basis of glycan modification in PD-1.** (**a**) A cartoon representation of the PD-1 structure. The four N-linked glycosylation sites (N49, N58, N74, and N116) are shown in sticks and coloured orange. The observed glycans are highlighted in sticks and coloured in magenta. (**b**) The 2 Fo-Fc electron density map of the N58 N-linked glycans contoured at 1.0 sigma is represented in blue. Three glycans consisting of two N-acetylglucosamine (NAGs) and one L-fucose (FUC) can be clearly observed. (**c**) SDS-PAGE analysis of the molecular reduction of four N-glycosylation single site-mutated PD-1 protein and PD-1 protein expressed in insect cells or refolded PD-1 protein expressed in *E. coli* cells. Compared with the WT PD-1 protein, each of the four N-glycosylation single site mutations has induced various levels of molecular weight reduction, indicating the N-glycosylation on each of the sites. PD-1 protein expressed in insect cells or *E. coli* cells showed substantial reduction of molecular weight, indicating the heavy glycosylation of PD-1.

and BC loop (Fig. 7a). Indeed, these four loops are very flexible, and showed different conformations when binding to different ligands (Fig. 7b). The FG loop of PD-1 shifted ∼6.9 Å away from that of the apo structure when binding to nivolumab, whereas the BC loop of PD-1 shifted ∼5.3 Å away when it binds to nivolumab or PD-L1. The C'D loop is so flexible that it could only be observed when stabilized with pembrolizumab and was not observed in crystal structures of apo PD-1 or complexes with PD-L1 or nivolumab.

From the PD-1-nivolumab-Fab and PD-1-pembrolizumab-Fab complex structures, the flexible loops within PD-1 were more likely to be targeted by therapeutic antibodies and they are promising targets for future drug development, especially structure-based drug design.

## Discussion

In this study, we report the structural basis of the nivolumab-based PD-1-PD-L1 blockade and N-glycosylation influences on PD-1-nivolumab interaction. Previous studies of PD-1-PD-L1 interactions or PD-1/PD-L1 targeting MAbs usually focused on the IgV domains of PD-1 and PD-L1 (refs 27,37–39). The PD-1 proteins used in these studies were obtained from the prokaryotic expression system and *in vitro* refolding. The present finding that the N-loop contributed major interactions with nivolumab is indeed unexpected. The N-loop of mPD-1 is disordered in the mPD-1/hPD-L1 complex, indicating its flexibility in ligand recognition[38]. The N-loop stabilized by nivolumab in the PD-1-nivolumab complex indicates that the flexible terminal loops outside the functional domain could also be valuable targets for drugs design. Though N-loop is the dominant contributor to the binding by nivolumab, it does not affect the interaction of PD-1-PD-L1, given that the N-loop is away from the binding surface of

PD-1-PD-L1 and that the binding affinity was not affected by the truncation of the N-loop. Na *et al.*[27] reported that the binding of PD-1 with pembrolizumab was located on the C'D loop of PD-1 and a site mutation (D85G) on the loop would completely abolish binding. The PD-L1-avelumab complex structure, as reported by our group, demonstrated that the FG loop of PD-L1 contributed major interactions with avelumab[26]. All these findings show that the loops were more likely to be targeted by MAbs in the immune checkpoint blockade of PD-1-PD-L1.

Immune checkpoint blockade therapies targeting PD-1, PD-L1 or CTLA-4 showed different levels of activities in clinical application likely due to different roles of the checkpoint molecules in tumour immunosuppression or progression[41–43]. Nivolumab and pembrolizumab are the only two MAbs available in clinical application with similar clinical efficiencies[25,44–46]. The recently reported PD-1-pembrolizumab-Fab complex structure enabled us to compare the binding mode and blockade mechanisms of these two antibodies[27]. These two antibodies showed a similar binding mode to PD-1, in which the dominant interaction was located on the loops of PD-1 and the competitive binding with PD-L1 utilizes steric clash to abolish the binding of PD-1-PD-L1. However, the targeted loops were completely different in that the pembrolizumab mainly binds to the C'D loop, whereas nivolumab mainly binds to the N-loop, with no overlapping binding areas on PD-1 with each other. Competitive binding analysis also reveals the partial complementary binding of nivolumab and pembrolizumab. Therefore, simultaneous administration of pembrolizumab and nivolumab may be considered for future therapy.

Even though the N-glycosylation on N58 was clearly observed in the structure located near the interface of PD-1-nivolumab-Fab complex, the absence of the N-glycan on N58 does not even affect the binding affinity to nivolumab, as indicated in the present

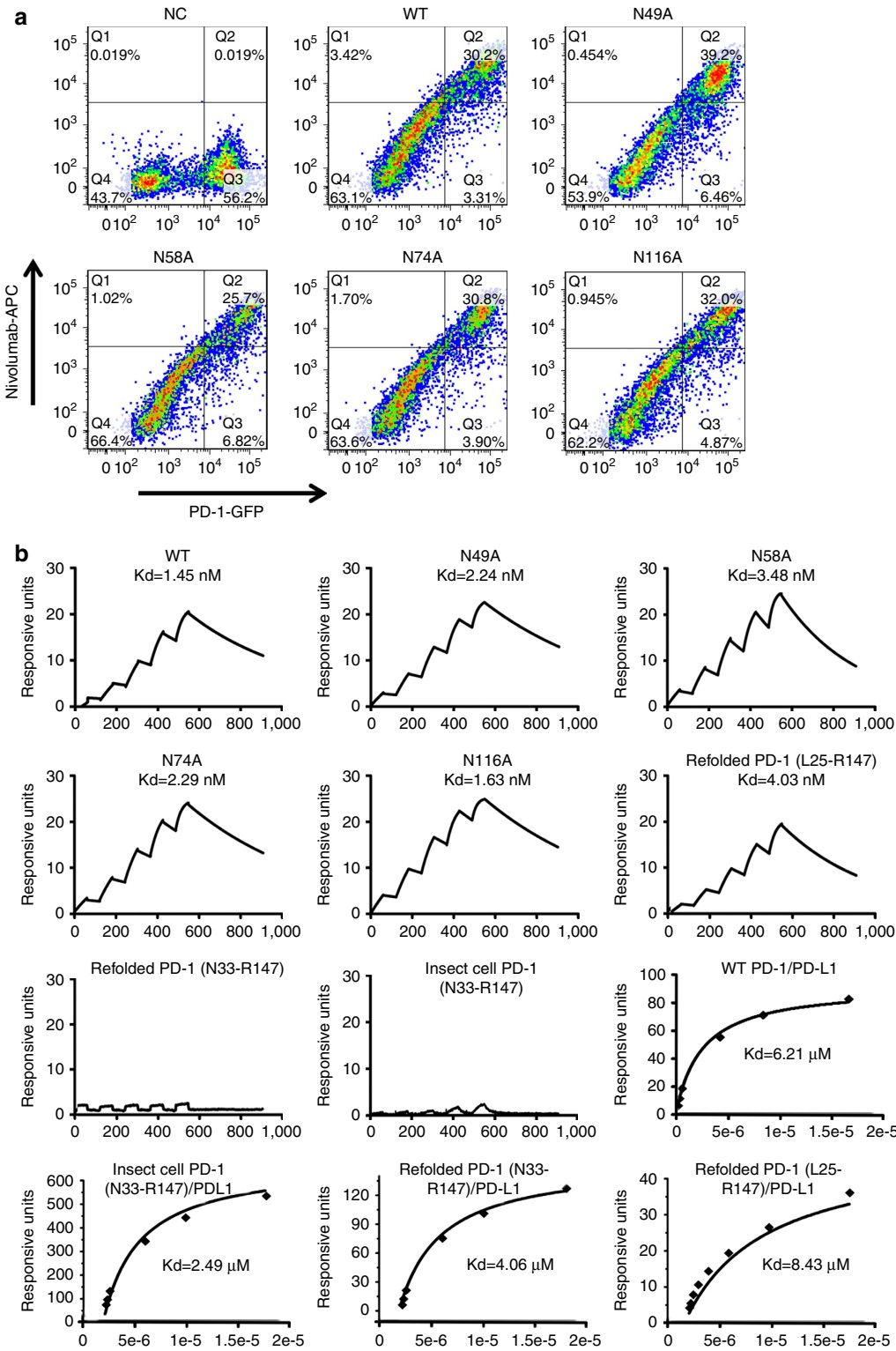

**Figure 4 | No evidence for N-glycosylation involvement in nivolumab recognition.** (**a**) A flow cytometric assay of nivolumab binding to PD-1 WT or various N-linked glycosylation sites mutated proteins (N49A, N58A, N74A and N116A) expressed on the cell surface of 293 T cells. Plasmids expressing the full-length PD-1 WT or mutant proteins fused with EGFP at the C terminus were used for transfection. Mock-transfected 293 T cells were used as negative control (NC). (**b**) SPR assay characterisation of the binding between nivolumab and various PD-1 mutant proteins using a BIAcoreT100 system. The PD-1 N-linked glycosylation sites mutant proteins expressed in 293 T cells were used for the assay. The N-linked glycosylation sites mutations in PD-1 showed no effect to the nivolumab binding. The refolded PD-1 proteins with (L25-R147) or without N-loop (N33-R147) expressed in E. coli and PD-1 protein without N-loop (N33-R147) from insect cells were then analysed for binding affinity with nivolumab. The roles of PD-1 N-loop to the binding of PD-L1 were analysed with response units plotted against protein concentrations. No substantial differences was detected among WT PD-1 expressed in 293 T cells, refolded PD-1 (L25-R147) expressed in E. coli, and N-loop truncated PD-1 (N33-R147) expressed in insect cells or E. coli. The data presented here are a representative of three independent experiments with similar results.

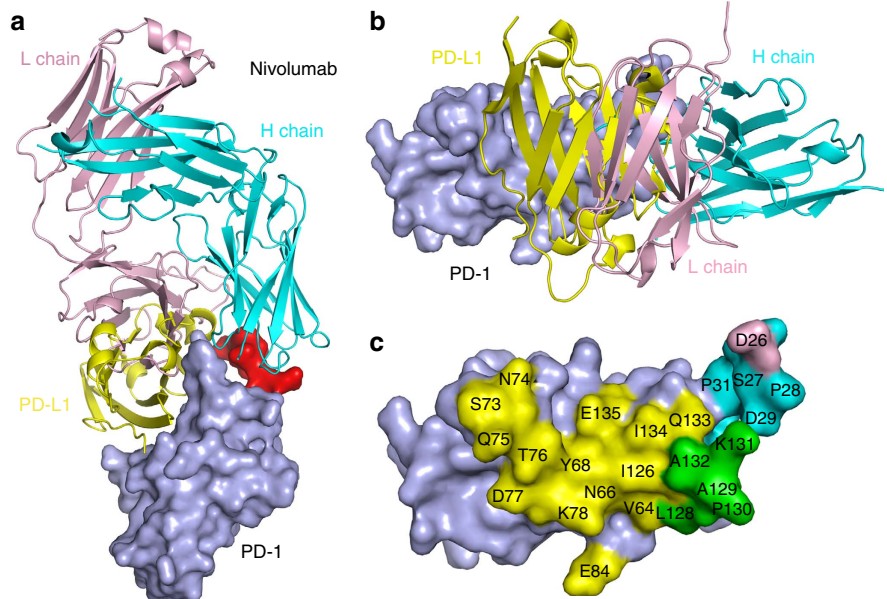

**Figure 5 | Competitive binding of nivolumab-Fab and PD-L1 with PD-1.** Superposition of the PD-1-nivolumab-Fab complex structure with PD-1-PD-L1 complex structure in side view (**a**) or top view (**b**). PD-L1 is shown in yellow and nivolumab-Fab H-chain in cyan, L-chain in pink. (**c**) Binding surface of PD-1 with PD-L1 or nivolumab. The residues in contact with PD-L1 are coloured in yellow, whereas residues in contact with nivolumab H-chain or L-chain are coloured in cyan or pink, respectively, and the overlapping residues bounded by both PD-L1 and nivolumab are coloured in green.

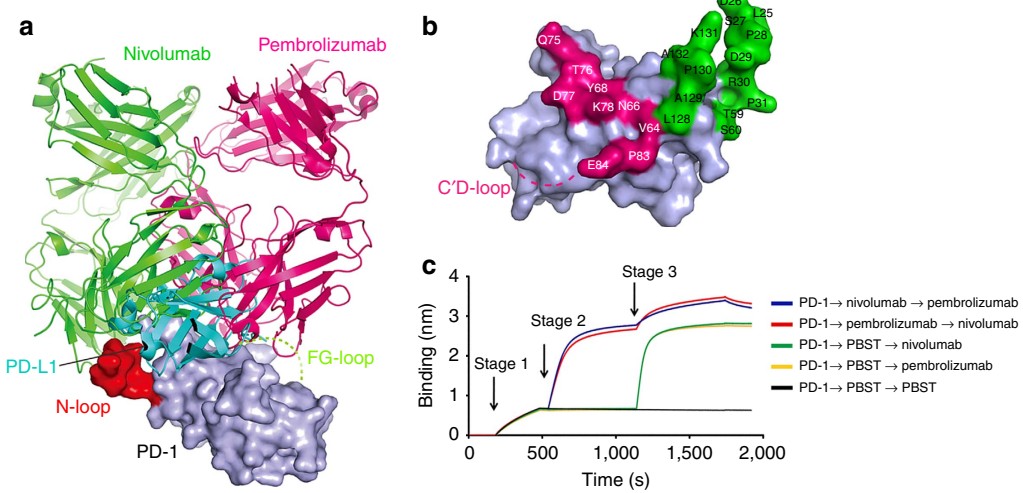

**Figure 6 | Distinct blockade binding mode compared with pembrolizumab.** (**a**) Superposition of PD-1-nivolumab-Fab, PD-1-pembrolizumab-Fab with the PD-1-PD-L1 complex structure. Nivolumab-Fab and pembrolizumab-Fab are coloured in green and magenta, respectively, and PD-L1 is coloured in cyan. PD-1 is shown as surface representation. The N-loop targeted by nivolumab is highlighted in red, and the unavailable pembrolizumab targeted C'D loop is indicated as dashed lines in lemon. Nivolumab and pembrolizumab bind to PD-1 in two different orientations with some clash regions. (**b**) Binding surface of PD-1 by nivolumab or pembrolizumab. The binding residues in contact with nivolumab are coloured in green, whereas residues in contact with pembrolizumab are coloured in magenta. (**c**) The Octet competition binding assay of nivolumab or pembrolizumab binding to PD-1. Ni-NTA sensors loaded with PD-1 at stage 1 were first saturated with the indicated MAbs (10 μg ml$^{-1}$) or PBST buffer at stage 2. The capacity of additional binding was monitored by measuring further shifts after injection of the other antibody (10 μg ml$^{-1}$) in the presence of the first antibody at stage 3. The result shows that nivolumab binding to PD-1 have partial competition with pembrolizumab.

study. Comparable binding affinity of nivolumab with PD-1 expressed either in mammalian cells or refolded PD-1 expressed in prokaryotic cells support that post-translational glycosylation does not affect the interaction of nivolumab with PD-1. Therefore, therapeutic efficacies of nivolumab would not be affected by the altered post-translational glycosylation especially when targeting intrinsic PD-1 in tumour cells[36]. The reason for previous report of nivolumab non-binding to *E. coli*-expressed

PD-1 is probably due to the use of the N-loop-truncated IgV protein for the assay(http://www.accessdata.fda.gov/drugsatfda_docs/nda/2014/125554Orig1s000PharmR.pdf).

In summary, we have reported the crystal structure of nivolumab-Fab in complex with the ectodomain of PD-1 expressed in mammalian cells. The observation that the N-loop other than the IgV domain of PD-1-dominated interactions with nivolumab was unexpected. The role of the N-loop in PD-1 MAb

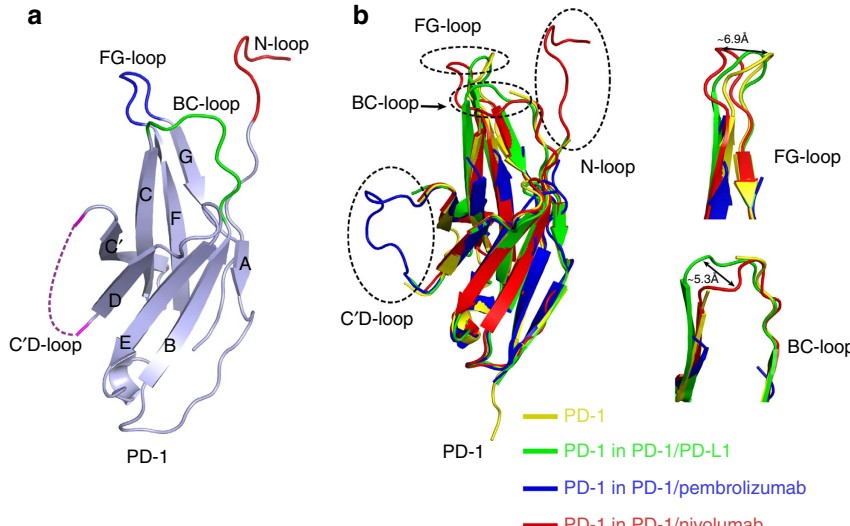

**Figure 7 | Targeting the loops of PD-1 for monoclonal antibody-based immune checkpoint blockade. (a)** The loops of PD-1 targeted by nivolumab or pembrolizumab are coloured differently as follows, N-loop, red; BC loop, green; FG loop, blue; C'D loop, magenta. **(b)** Comparison of PD-1 with other solved PD-1 structures with/without ligands (apo structure, yellow, PDB: 3RRQ; PD-1/PD-L1 structure, green, PDB: 4ZQK; PD-1-pembrolizumab-Fab structure, blue, PDB: 5JXE; PD-1-nivolumab-Fab structure, red).

development has been neglected, as the IgV construct has been widely used in the field. Glycosylation-independent interaction with nivolumab indicates a broader application of nivolumab in management of multiple tumours especially considering that the intrinsic expression of PD-1 in tumour cells is usually accompanied with altered glycan modifications. The N-loop dominated binding to nivolumab indicates that the flexible N-loop of PD-1 deserves more attention for future PD-1-PD-L1 blockade antibody design. Together with previous findings of loop-dominating interactions in PD-1-pembrolizumab and PD-L1-avelumab, targeting the PD-1/PD-L1 loops should be considered in future development of blockade drugs.

## Methods

**Plasmid construction and protein purification.** The DNA encoding the ectodomain of hPD-1 (residues M1-Q167, including signal peptide, UniProt: Q15116) with six histidines at the C terminus of the sequence was cloned into pCAGGS vector (Addgene) with *Eco*RI and *Bgl*II restriction sites. Plasmid pCAGGS-PD-1 was transiently transfected into 293 T cells (ATCC) for protein expression. To allow expression of the hPD-1 N-linked glycosylation mutant proteins, the codes substitutions of residues N49A, N58A, N74A or N116A were done by site-directed mutagenesis using the overlap extension PCR method (Supplementary Table 2) and expressed in the same way. The PD-1 coding fragment (N33-R147) was inserted into pFastbac1 vector (Invitrogen), in-frame with an N-terminal gp67 signal peptide for secretion and a C-terminal hexa-His tag for purification[46]. Transfection and virus amplification were performed according to the Bac-to-Bac baculovirus expression system manual (Invitrogen)[47,48]. The recombinant baculovirus was then used to infect Hi5 cells (Invitrogen) to produce soluble PD-1. The coding sequences for $V_H$ and $V_L$ of nivolumab-Fab were obtained from IMGT database (INN:9623) and codon optimized and synthesized by Genewiz Inc (Supplementary Table 3) and cloned into pFastBac-Dual vector (Invitrogen), which contains two multiple cloning sites to allow simultaneous expression of two heterologous genes in a single recombinant baculovirus under two different promoters, the polyhedrin promoter and p10 promoter. The $V_H$ DNA was cloned into the *Sma*I to *Kpn*I sites under control of p10 promoter, with an N-terminal honeybee melittin signal peptide for protein secretion and a C-terminal hexa-His tag to facilitate further purification processes. The $V_L$ DNA was sub-cloned into the *Bam*H I to *Hind* III sites under the control of polyhedrin promoter, with an N-terminal gp67 signal peptide and a C-terminal flag tag for detection of expression. Protein was then prepared with the Bac-to-Bac baculovirus expression system as above. In each case, the supernatant was collected and the protein was purified by sequentially His-Trap HP column (GE Healthcare) and Superdex 200 10/300 GL(GE Healthcare) in a buffer containing 20 mM Tris and 150 mM NaCl (pH 8.0).

Expression of the full-length nivolumab or pembrolizumab used for functional analysis was achieved by co-transfection of two plasmids, one encoding the $V_L$ and the other one the $V_H$, into 293 T cells. The protein was purified from the culture supernatants using a Protein G affinity column (GE Healthcare) and subsequently purified by gel filtration on a Superdex 200 10/300 GL(GE Healthcare) in a buffer containing 20 mM Tris and 150 mM NaCl (pH 8.0).

Two constructs encoding the PD-1 fragments (residues L25-R147 or N33-R147) were separately cloned into the pET-21a vector (Novagen) with *Nde*I and *Xho*I restriction sites and transformed into *E. coli* strain BL21 (DE3) for protein expression. PD-1 was over expressed in *Escherichia coli* as inclusion bodies and subsequently refolded *in vitro*[26]. In brief, the dissolved PD-1 inclusion body was diluted against a refolding buffer (100 mM Tris, pH 8.0; 400 mM L-Arginine; 5 mM EDTA-Na; 5 mM Glutathione; 0.5 mM Glutathione disulfide). After 12 h of slow stirring at 4 °C, the refolded PD-1 was then concentrated and changed to a 20 mM Tris-HCl (pH 8.0) and 150 mM NaCl buffer and further analysed by Superdex 200 10/300 GL (GE Healthcare) chromatography. The PD-L1 protein (residues F19-R238, UniProt: Q9NZQ7) was cloned into the pET-21a vector (Novagen) with *Nde*I and *Xho*I restriction sites and prepared like PD-1 protein as described above.

**Complex preparation and crystallisation.** The mammalian cell expressed PD-1 protein and nivolumab-Fab fragment were mixed at a molar ratio of 1:1. The mixed sample was incubated for 30 min on ice and then purified by gel filtration. 10 mg ml$^{-1}$ of pooled proteins were used for crystal screening by vapour-diffusion sitting-drop method at 4 °C. Diffracting crystals were obtained in a condition consisting of 0.2 M ammonium acetate, 0.1 M BIS-TRIS pH5.5, 25% w/v poly-ethylene glycol 3,350.

**Data collection and structure determination.** After incubation in a reservoir solution containing 20% (v/v) glycerol crystals were flash-cooled in liquid nitrogen. The diffraction data were collected at Shanghai Synchrotron Radiation Facility BL17U, and all data were processed with HKL2000 (ref. 49). The complex structure was solved by molecular replacement method using Phaser[50,51] from the CCP4 programme suite[52], with the reported PD-1 structure (PDB: 3RRQ) and Fab structure (PDB: 3EYQ) as the search models. COOT[53] and PHENIX[54] were used for subsequent model building and refinement. The stereochemical qualities of the final model were assessed with MolProbity[55]. Data collection and refinement statistics are summarized in Table 1. All structure figures were prepared with Pymol (http://www.pymol.org).

**N-terminal sequencing.** The mammalian cell expressed PD-1 protein samples were first applied to SDS-PAGE and then transferred to a polyvinylidene fluoride (PVDF) membrane. The PVDF blot membrane was stained and destained until the protein bands became visible[46]. The PVDF membrane was dried and the band of PD-1 was cut for the N-terminal sequencing with the Edman degradation method using PROCISE491 (America Applied Biosystems).

**FACS analysis of nivolumab binding to PD-1 mutants.** To obtain cell surface-expressing PD-1 fused with EGFP protein, the full-length PD-1 was cloned into the pEGFP-N1 vector (Clontech). The plasmids expressing the PD-1 mutants N49A, N58A, N74A or N116A were done by site-directed mutagenesis (Supplementary Table 2). The plasmids were transfected into 293 T cells, respectively. Cells were collected 48 h after transfection and resuspended in PBS at $1 \times 10^7$ cells ml$^{-1}$ Then the 293 T cells expressing wild-type PD-1 or mutants were stained with nivolumabat room temperature for 30 min, washed three times with PBS and further stained with secondary APC-anti-human IgG (Clone G18-145, BD Pharmingen) for another 30 min. Cells were analysed by flow cytometry with a BD FACS Aria II machine after washing.

**SPR analysis.** SPR measurements were done at room temperature using a BIAcoreT100 system with CM5 chips (GE Healthcare). For all measurements, an HBS-EP buffer consisting of 150 mM NaCl, 10 mM HEPES, pH 7.4 and 0.005% (v/v) Tween-20 was used as running buffer, and all proteins were exchanged into this buffer in advance through gel filtration. The blank channel of the chip served as the negative control. To detect the nivolumab binding to different forms of PD-1 proteins, full-length nivolumab was immobilized on the chip at a concentration of 1 µg ml$^{-1}$ by anti human IgG at ∼70 response units. Gradient concentrations of PD-1(0 nM, 3.125 nM, 6.25 nM, 12.5 nM, 25 nM and 50 nM) were then flowed over the chip surface. After each cycle, the sensor surface was regenerated with 3 M MgCl$_2$.For the binding between PD-1 and PD-L1 detection, PD-1 was immobilized on the chip. The binding kinetics were all analysed with the software of BIA evaluation Version 4.1 using a 1:1 Langmuir-binding model.

**Binding competition assay.** Binding competition between nivolumab and pembrolizumab was determined using a real-time, label-free bio-layer interferometry assay on an Octet RED96 biosensor (Pall ForteBio). The entire experiment was done at 30 °C in PBS with Tween (PBST) buffer with the plate shaking at a speed of 1,000 rpm. Ni-NTA biosensors (Pall ForteBio) were first loaded with 10 µg ml$^{-1}$ PD-1 protein for 300 s and then associated with first MAbs(10 µg ml$^{-1}$) for 600 s to get saturation. Afterwards, the biosensors were dipped into the second MAb (10 µg ml$^{-1}$) in the presence of the first one for another 600 s. PBST buffer was used as negative control. All MAbs were evaluated at concentration of 10 µg ml$^{-1}$. All sensors were regenerated with 10 mM Glycine-HCl (pH 1.7, GE Healthcare) and re-charging with 10 mM NiCl$_2$. The real-time binding response was measured during the course of the experiment. The responses of the two MAbs binding to PD-1 were compared and competitive/noncompetitive behaviour was determined.

**Data availability.** Coordinates and structure factor of the structure reported here have been deposited into the Protein Data Bank with PDB Code: 5WT9. The PDB accession codes 4ZQK, 3RRQ, 5JXE were used in this study. The UniProt accession codes Q15116, Q9NZQ7 were used in this study. All other relevant data are available from the corresponding authors upon reasonable request.

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

## Acknowledgements

This work was supported by the National Natural Science Foundation of China (NSFC; 31390432 and 31500722) and the External Cooperation Program of Chinese Academy of Sciences (Grant No. 153211KYSB20160001). We would like to thank the staff at the Shanghai Synchrotron Radiation Facility (SSRF-beamline 17U) for their assistance in diffraction data collection. We also thank Yuanyuan Chen and Zhenwei Yang from Institute of Biophysics Chinese Academy of Sciences for their technical support in the SPR assay. Q.W. is supported by Young Elite Scientist Sponsorship Programme by CAST (YESS). G.W. is the recipient of a Banting Postdoctoral Fellowship from the Canadian Institutes of Health Research (CIHR) and the President's International Fellowship Initiative from the Chinese Academy of Sciences (CAS).G.F.G. and J.Y. are supported by the NSFC Innovative Research Group (Grant No. 81621091).

## Author contributions

G.F.G., J.Y. and S.T. designed and supervised the study. H.Z, S.T., Z.T., J.W., C.W. and X.Z. conducted the experiments. Y.C., H.S. and J.Q. collected the data sets and solved the structures. S.T., H.S., Q.W., G.W., W.L., S.G., Z.W., Y.S., F.Y., G.F.G. and J.Y. analysed the data and wrote the manuscript.

## Additional information

**Competing financial interests:** The authors declare no competing financial interest.

