## [Peer review file · Nature Communications]

Reviewers' comments:

Reviewer #1 (Remarks to the Author):

Modulating immune checkpoint pathways like CTLA-4 and PD-1 for cancer immunotherapy has gained much attention in recent years due to their immense potential for treating malignancies. In this very straightforward paper, the authors report the crystal structure of PD-1 in complex with nivolumab, one of the two clinically approved anti-PD-1 antibodies. The structure, combined with mutational analysis, reveals a critical role for an N-terminal loop (N-loop) of PD-1 in binding nivolumab. Although the N-loop is not involved in binding PD-L1 or pembrolizumab, the epitope on PD-1 recognized by nivolumab partially overlaps the PD-L1 binding site, which provides a mechanism for PD-1/PD-L1 blockade by this antibody. The authors also compare the binding mode of nivolumab with pembrolizumab (the other clinically approved anti-PD-1 antibody), whose structure with PD-1 was solved recently, and point out that the binding epitopes of the two antibodies are different, suggesting the possibility of simultaneous administration of them both, should any escape mutants arise in the future. The authors evaluate the effects of glycosylation and demonstrate that the interaction of nivolumab with PD-1 is clearly glycosylation-independent. This is a useful finding since nivolumab can potentially be administered without altering efficacy in tumor cells with glycan modifications. The structural results reported here, along with previous information on the PD-1/pembrolizumab complex, may facilitate development of small molecule inhibitors of the PD-1/PD-L1 interaction. This study will be of considerable interest to the rapidly expanding checkpoint blockade community.

Points to address:

1. The authors presumably obtained the sequence of nivolumab from the patent literature. They should cite this patent.
2. Calling the N-loop “long-neglected” is not entirely accurate. Although most PD-1 constructs used for structure determinations start at residue 32, 33 or 34, thereby excluding the N-loop, there are two exceptions. The first is the PD-1 construct reported here, which begins at residue 25. The second is a PD-1 construct used for structure determination of the mouse PD-1/PD-L1 complex (PDB accession code 3BIK), which also begins at residue 25. However, the N-loop is disordered in the mouse PD-1/PD-L1 complex, indicating flexibility. Nivolumab evidently stabilizes a particular conformation of the flexible N-loop, thus allowing it to be seen. The authors should clarify this point.
3. Regarding the CDER report (ref. 36) that nivolumab did not bind nonglycosylated PD-1 expressed in *E. coli*, this is likely because the construct used by BMS did not include the N-loop. Although details of that PD-1 construct are not provided in ref. 36, the authors should comment in the Discussion that the (presumed) absence of the N-loop in the BMS *E. coli* PD-1 construct probably explains why this construct did not bind nivolumab. This would explain the difference in binding results and avoid possible confusion on the part of the reader.
4. Line 268, page 14: “nivolumab” instead of “avelumab”
5. Figure 1b: The peak labeled “survived” should be labeled “complex”

Reviewer #2 (Remarks to the Author):

The manuscript reports the crystal structure of nivolumab binding to PD-1, pinpointing the binding of nivolumab to a N-terminal loop outside the IgV domain of PD-1, which is not required for PD-L1 binding. Furthermore it is shown that nivolumab binding is not dependent on glycosylation of PD-1 and that nivolumab has a different interaction with PD-1 than pembrolizumab has.

Questions and remarks:

1. This type of study is of interest given the current attention for and success of immune checkpoint blockade therapy to treat cancer and could support drug design. The question however is how the identification of this “neglected” loop as a binding site for nivolumab would influence design of novel reagents. This loop is not involved in PD-L1 binding, in stead the authors propose that “..the blockade mechanism of nivolumab is that the VL of nivolumab provides steric clash to abrogate the binding of PD-L1 to PD-1..”. Give this finding, the proposed binding site will most likely not be of interest to target with i.e. small molecule drugs.
2. Since PDL1 and PDL2 are interacting in a structurally different way (Lazar-Molnar et al PNAS 2008), it would be important to know whether the proposed binding N-loop is involved in PDL-2 binding.
3. In Figure 4 different glycosylation mutants are expressed in 293T cells and compared for binding of nivolumab, but not for PD1-L binding. The conclusion that the N-loop is important for nivolumab binding however, is only drawn on the basis of refolded or insect cell expressed PD-1 and PD-1 mutants. An analysis of PDL-1 and nivolumab binding to a 293T expressed N-loop mutant of PD-1 would have been a better comparison to assess the importance of the N-loop in both PD-1 and nivolumab binding in a relevant setting. This would also further clarify apparent differences of the current data with previous reports regarding the requirement of N-linked glycosylation for both the binding of PD-1 ligands as nivolumab to PD-1.
3. Although tumor intrinsic PD-1 expression has been reported for melanomas, the main effect of PD-1 blocking antibodies is assumed to be by unleashing T cell activity. Whereas tumors may escape antibody therapy by outgrowth of tumor cells that escape antibody binding, it is unlikely that T cells will be selected against PD-1 antibody binding. It is therefore to be questioned how relevant the possibility of therapy resistance by mutation of PD-1 on T cells is and this should be discussed accordingly.
4. The English language use in the paper needs extensive editing

Reviewers' comments:

Reviewer #1 (Remarks to the Author):

Points to address:

1. The authors presumably obtained the sequence of nivolumab from the patent literature. They should cite this patent.

Reply: The sequence of nivolumab was obtained from the IMGT data base and we have cited this in line 313.

2. **Calling the N-loop “long-neglected” is not entirely accurate.** Although most PD-1 constructs used for structure determinations start at residue 32, 33 or 34, thereby excluding the N-loop, there are two exceptions. The first is the PD-1 construct reported here, which begins at residue 25. The second is a PD-1 construct used for structure determination of the mouse PD-1/PD-L1 complex (PDB accession code 3BIK), which also begins at residue 25. However, the N-loop is disordered in the mouse PD-1/PD-L1 complex, indicating flexibility. Nivolumab evidently stabilizes a particular conformation of the flexible N-loop, thus allowing it to be seen. **The authors should clarify this point.**

Reply: We agree that the “long-neglected” is overemphasized and should be removed. We have made adjustments in lines 39, 156 and 244. The flexibility of N-loop and its implication was further discussed in line 244-248 as “The N-loop of murine PD-1 (mPD-1) is disordered in the mPD-1/human PD-L1 complex, indicating its flexibility in ligand recognition³⁹. The stabilized N-loop in nivolumab/PD-1 complex by nivolumab indicates that the flexible terminal loops outside functional domain could also be valuable targets for drug design.”

3. Regarding the CDER report (ref. 36) that nivolumab did not bind nonglycosylated PD-1 expressed in E. coli, this is likely because the construct used by BMS did not include the N-loop. Although details of that PD-1 construct are not provided in ref. 36,

the authors should comment in the Discussion that the (presumed) absence of the N-loop in the BMS E. coli PD-1 construct probably explains why this construct did not bind nivolumab. This would explain the difference in binding results and avoid possible confusion on the part of the reader.

Reply: We mentioned this in line 281-283 that “The reason for previous record of nivolumab non-binding to E. coli-expressed PD-1 is probably due to the use of N-loop-truncated IgV protein for the assay³⁷.”. We believe this will avoid possible confusion to the reader.

4. Line 268, page 14: “nivolumab” instead of “avelumab”

Reply: We are sorry for our careless mistake and this has now been changed in line 271.

5. Figure 1b: The peak labeled “survived” should be labeled “complex”

Reply: The label was changed as suggested in the new Figure 1b.

Reviewer #2 (Remarks to the Author):

Questions and remarks:

1. This type of study is of interest given the current attention for and success of immune checkpoint blockade therapy to treat cancer and could support drug design. The question however is how the identification of this “neglected” loop as a binding site for nivolumab would influence design of novel reagents. This loop is not involved in PD-L1 binding, instead the authors propose that “..the blockade mechanism of nivolumab is that the VL of nivolumab provides steric clash to abrogate the binding of PD-L1 to PD-1.”. Give this finding, the proposed binding site will most likely not be of interest to target with i.e. small molecule drugs.

Reply: We agree that the finding of the N-loop dominated interaction with

nivolumab in the present study will likely not be of interest for small molecule drug design. Actually, we tried to propose that the flexible N-loop provides an important epitope and deserve more attention for future PD-1/PD-L1 blockade antibody design, despite the fact that it is outside the Ig domain of PD-1. We have adjusted our comments in line 244-248 as answered to Reviewer 1 above and the conclusive comments have been changed accordingly in the last part of the discussion (line 290-292), as “The N-loop dominated binding to nivolumab indicated that the flexible N-loop of PD-1 deserve more attention for future PD-1/PD-L1 blockade antibody design.”.

2. Since PDL1 and PDL2 are interacting in a structurally different way (Lazar-Molnar et al PNAS 2008), it would be important to know whether the proposed binding N-loop is involved in PDL-2 binding.

Reply: The structure of human PD-1/PD-L2 complex has not yet been solved, but the binding mode may be similar to the murine PD-1/PD-L2. The N-loop of PD-1 was also absent in the interaction of mPD-1 with mPD-L2 as it has been observed in the mPD-1/hPD-L1 complex. While there are possible uncertainties within hPD-1/hPD-L2, we prefer not to discuss this issue before hPD-1/hPD-L2 complex structure is determined, on which we are working now.

3. In Figure 4 different glycosylation mutants are expressed in 293T cells and compared for binding of nivolumab, but not for PD1-L binding. The conclusion that the N-loop is important for nivolumab binding however, is only drawn on the basis of refolded or insect cell expressed PD-1 and PD-1 mutants. An analysis of PDL-1 and nivolumab binding to a 293T expressed N-loop mutant of PD-1 would have been a better comparison to assess the importance of the N-loop in both PD-1 and nivolumab binding in a relevant setting.

This would also further clarify apparent differences of the current data with previous reports regarding the requirement of N-linked glycosylation for both the binding of PD-1 ligands as nivolumab to PD-1.

Reply: We agree that additional binding experiments may be better to support N-loop dominated binding to nivolumab. Our results revealed that the PD-1 (L25-R147) expressed from 293T cell and refolded PD-1 (L25-R147) showed comparable binding affinities to both PD-L1 (Kd=6.21 μ M and 8.43 μ M, respectively) and nivolumab (Kd=1.45 nM and 4.03 nM, respectively), which indicates that the binding of PD-1 to both PD-L1 and nivolumab was completely irrelevant to the protein expression system. The absence of binding of the N-loop truncated PD-1 obtained from refolding or insect cells to nivolumab has provided sufficient evidence that the lack of N-loop in PD-1 would abrogate the interaction with nivolumab while no substantial influences to PD-L1 binding (Kd=4.06 μ M and 2.49 μ M, respectively) was observed.

Due to the complete lack of post-translational glycosylation of *E.coli* expressed proteins, the comparable binding affinities between PD-1 (L25-R147) expressed from 293T cell and refolded PD-1 (L25-R147) (Kd=1.45 nM and 4.03 nM, respectively), together with the N-glycan site mutations, to nivolumab clearly showed that the interaction between PD-1 and nivolumab is completely independent of glycosylation.

Therefore, the reviewer's proposed experiments could be deduced from the current data and our conclusions are justified with the available data reported here.

4. Although tumor intrinsic PD-1 expression has been reported for melanomas, the main effect of PD-1 blocking antibodies is assumed to be by unleashing T cell activity. Whereas tumors may escape antibody therapy by outgrowth of tumor cells that escape antibody binding, it is unlikely that T cells will be selected against PD-1 antibody binding. It is therefore to be questioned how relevant the possibility of therapy resistance by mutation of PD-1 on T cells is and this should be discussed accordingly.

Reply: Cases of ineffective PD-1 targeting antibody treatment have indeed been observed in clinical trials (Herbst et al., 2014; Tumeh et al., 2014; Rizvi et al., 2015). PD-L1 expression levels, compensative expression of other inhibitory

molecules, pathological types of tumors, the stage of disease, somatic mutation levels and the resulting abundance of neoantigens, and the number of TILs, etc. are all factors in determining the efficacy of immunotherapy. We do agree with this point that antibody driven therapy resistance by PD-1 mutation is less possible on T cells and thus discussions referring to MAb driven PD-1 mutation are removed in line 117 and 271.

5. The English language use in the paper needs extensive editing

Reply: The manuscript has been extensively edited for grammar and clarity by a native English speaker (Gary Wong), who is part of the authors list. We will coordinate with the staff at *Nature Communications* to resolve any further issues.

REVIEWERS' COMMENTS:

Reviewer #1 (Remarks to the Author):

The authors have satisfactorily addressed the previous reviewers' critiques.